# Sustainable Transportation for Events: A Systematic Review

**Dario Ballarano [1],\*, Sergio Maria Patella [2] and Francesco Asdrubali [1]**

1   Department of Industrial, Electronic and Mechanical Engineering, Roma Tre University, 00146 Rome, Italy
2   Faculty of Economics, Universitas Mercatorum, 00186 Rome, Italy
\*   Correspondence: dario.ballarano@uniroma3.it

**Abstract:** Event business is constantly growing in frequency and size, attracting people from nearby areas and different countries as well. Transportation plays a key role in a successful event, especially for major ones, where the large number of participants requires ad hoc designing of event mobility and might include implementation of new infrastructures. In recent decades, environmental issues have emphasized event negative externalities, leading to the design of green events. This paper presents a comprehensive systematic literature review on studies investigating the sustainable practices related to event mobility. The methodology showcases a selection of 32 articles, which were classified according to their main contributions into five categories, for which gaps and trends were shown. Our findings indicate that there are not enough studies to draw conclusions about good practices that can be generalized to each kind of event. Additionally, strong limitations in the reviewed papers are the different contexts of the analysis and the lack of a validation through empirical data. The research presented in this paper adds a contribution to the sustainable event transportation literature reporting the most established models, methods, and interventions. In detail, the most promising interventions involve mass transit and demand-oriented services that rely on intelligent transportation systems and user information.

**Keywords:** large events; green mobility; sustainable transportation; literature review

## 1. Introduction

One of the first definitions of "sustainable development" was provided by the World Commission on Environment and Development, which defined it as "the development that meets the needs of the present without compromising the ability of future generations to meet their own needs" [1]. The United Nations (UN) 2030 Agenda established a framework for sustainable development summarized in 17 Sustainable Development Goals (SDGs), which include a complex of attainable objectives, meaningful actions, publications, events, and the general promotion of a proactive sustainability-oriented behavior, among all UN member states [2]. In the European community, these directives have been incorporated to address climate change and environmental degradation challenges. In this regard, one-third of the 1.8 trillion EUR of investment planned by the Next Generation EU recovery plan and, in equal measure, by the EU's 7-year budget, will be allocated to the European Green Deal, consisting of a series of proposals in order to transform EU climate, energy, transport, and taxation policies to reduce net greenhouse gas emissions by at least 55% by 2030 compared to 1990 levels [3]. An important role in sustainable development is played by the transportation sector. As reported by IEA [4], greenhouse gas (GHG) emissions from road transport represent 24% of overall global direct $CO_2$ emissions. Daily life activities are facilitated and positively influenced by improvements in the efficiency of transportation systems; among relevant effects, a reduction in travel times and an overall increase in productivity have been found [5]. Urban development and transportation are often perceived as slow evolutionary processes, yet there are some opportunities, such as mega-events, to transform cities' transportation systems in just 10 years or less [6]. Mega-events attract hundreds of thousands of attendees and can be seen as a catalyst for urban

development of a metropolis. This is an opportunity to assert its status as a global city, which is why mega-events are given a prominent position in urban agendas of modern cities [7]. The challenge in using mega-event opportunities is to align the transportation requirements for hosting a world-class event with the metropolitan vision, using mega-events as a tool for desired changes. Indeed, in addition to the importance of preparing transportation systems for major events, these preparations also serve as tangible legacies that can improve urban transportation by encouraging a change in people's travel habits [6]. Allen et al. (2005) and Getz (2008) [8,9] classified events according to their size, shape, type of visitors, and frequency. They gave the following definition: Events are celebrations of unique occasions that are organized to achieve certain goals and can be classified according to their size, form, type of visitors, and frequency. Events size varies from local/community events to mega-events, and the size is directly related to the impact [10]. In fact, large-scale events are more frequently the target of advertisements, and, in recent years, they have become increasingly significant in driving travel demand and contributing to increasing GHG emissions [11]. In this context, the Event Industry Council (EIC), a group of event industry leaders from around the world which recognize the importance of sustainability and support it, was developed [12]. A definition of sustainability for events is provided by Events Council [13], which defines four main guidelines to create a sustainable event:

(1) Event organizers and suppliers share responsibility for implementing and communicating sustainable practices to their stakeholders;
(2) Basic environmental practices (resource conservation, including waste management, carbon emissions reduction and management, supply chain management, ethical consumption, and biodiversity conservation);
(3) Basic social considerations (universal human rights, community impact, labor practices, respect for culture, safety and security, and health and wellbeing);
(4) Sustainable events support thriving economic practices (collaboration and partnerships, local support particularly for small and medium-sized enterprises (SMEs), stakeholder engagement, equal economic impact, transparency, and responsible governance).

Similarly, the EIC seeks to incorporate the SDGs into the event sector by adhering to seven accepted criteria for events, each representing a different area of activity. The EIC standards include event planner, accommodations, audio and video production, destination, exhibition, catering, and event venue [12]. Several efforts have been made to fill the transportation gap, and several branches of research are proposed in the literature: promotion of guidelines and best practices [12,14]; development of environmental impact assessment models [15–17]; development of transportation simulation models [18–21]; analysis of spectator behavior and satisfaction [22,23]; analysis of previous cases and their legacy [7,19,24–27]. Public transportation [28–30], facility location planning [31], active mobility and ridesharing [12], use of alternative fuels [28], and information technologies [7] seem to be the most effective practices to pursue sustainability. Special attention is paid to active mobility, with bike-sharing systems (BSSs) being widely regarded as a pro-environmental approach compared to motorized modes [29]. The complexity in designing sustainable mobility for an event follows from its inherent characteristics: The different demand and supply characteristics in the pre-event, during, and post-event phases that affect the local transportation system [7]. According to Kassens (2009) [6], it is crucial to learn from past case studies to avoid mistakes in designing and collecting effective and potential benefits; nonetheless, studies on event-related sustainable transportation are quite rare. Sustainable transport is defined as one of the main challenges; however, different approaches, simplified models, and numerous approximations highlight that it has not been treated as a main focus in the tourism destination and event management literature [32]. Research in this area is immature, and there are no technical review documents. Additionally, there are only papers dealing with short literature reviews, which are mainly addressed to the sustainability of the event in general rather than to transportation. This review collects the contributions of authors and reports the state of research on sustainable mobility for events providing greater accessibility and visibility of results, approaches, and

models. This is performed through a systematic analysis of the bibliography based on sets of keywords used in different major search engines. The research presented in this paper highlights the lack of studies in a sector that is constantly growing and generates numerous externalities. Moreover, this study indicates the most effective practices for pursuing sustainability in event transportation. Lastly, five recurring themes are identified, and the temporal evolution of scientific interest in each topic is assessed.

The paper is organized as follows. Section 2 outlines the bibliographic research methodology, while results are presented in Section 3. Section 4 concludes the paper and discusses directions for future research.

## 2. Materials and Methods

Inspired by Denyer et al. (2009), Kilibarda et al. (2020), and Patella et al. (2020) [33–35], the literature selection procedure is shown in Figure 1. The planning phase, identification phase, screening phase, and selection phase can be distinguished. The planning phase consists of the identification of gaps, the aim of the research, and the need for the present review. Three main research questions were set (RQ1, RQ2, and RQ3):

- RQ1: What are the main research directions and trends on sustainable event transportation?
- RQ2: Which are the most effective sustainable transportation practices?
- RQ3: What are the most appropriate models and methods for quantifying mobility effects in events on the environment?

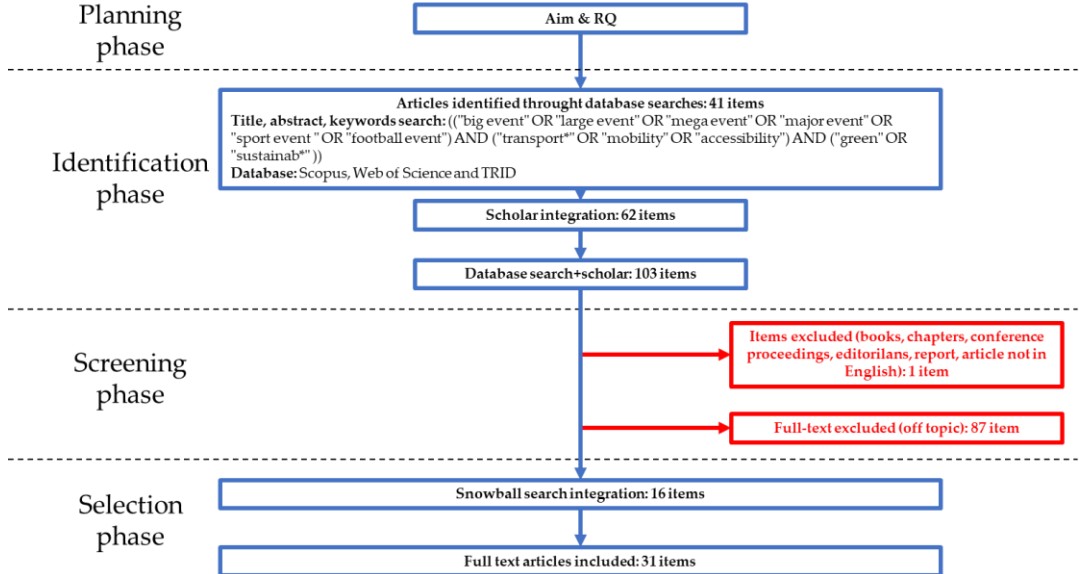

**Figure 1.** Literature selection procedure.

The aim is to identify and gather research by focus trends and to analyze the evolution of scientific interest over time. The second research question aims to collect and return the most promising sustainable practices in event transportation. The third research question investigates the context of the approaches and types of models and methods most established in the literature.

After the completion of the research planning, we proceeded to the identification phase, which involved the definition of a set of search keywords and the relationships between them. The intention was to cover a wide range of information, in order to reduce bias and ensure the objectivity and validity of the research. It was carried out on the Scopus, Google Scholar, Web of Science, and TRID databases considering the set combination of the following keywords: "big event", "large event", "mega event", "major event", "transport*", "mobility", "accessibility", "green", and "sustainab*". In detail, the query string used to search across abstract, title, and keywords was TITLE-ABS-KEY (("big event"

OR "large event" OR "mega event" OR "major event") AND ("transport*" OR "mobility" OR "accessibility") AND ("green" OR "sustainab*")). The term "*" allows a wildcard search, which maximizes the search results. To ensure an adequate and homogeneous quality level, it was decided to include only English-language articles, editorials, and book chapters published in peer-reviewed journals without considering time constraints.

The identification phase collected search results from Scopus, Google Scholar, Web of Science, and TRID databases. In detail, the results from the Scopus, Web of Science, and TRID search yielded 41 unique articles, while those from the Google Scholar search yielded 317 articles. Below, the search keyword combinations matrix on Google Scholar and the number of results found for each are displayed. In detail, Table 1 collects all the keyword combinations obtained by the query string.

**Table 1.** Google Scholar search result.

|  | **Big Event** | **Large Event** | **Mega Event** | **Major Event** | **Sport Event** |  |
|---|---|---|---|---|---|---|
| Transport | 0 | 1 | 0 | 14 | 2 |  |
| Transportation | 0 | 2 | 0 | 12 | 2 | Green |
| Mobility | 1 | 2 | 0 | 7 | 1 |  |
| Accessibility | 1 | 1 | 0 | 6 | 0 |  |
| Transport | 1 | 4 | 2 | 14 | 7 |  |
| Transportation | 2 | 5 | 2 | 13 | 8 | Sustainability |
| Mobility | 3 | 2 | 1 | 4 | 1 |  |
| Accessibility | 0 | 2 | 2 | 9 | 5 |  |
| Transport | 2 | 12 | 3 | 35 | 8 |  |
| Transportation | 4 | 18 | 2 | 57 | 8 | Sustainable |
| Mobility | 5 | 6 | 1 | 16 | 1 |  |
| Accessibility | 0 | 0 | 0 | 0 | 0 |  |

Therefore, 255 duplicate articles in the Google Scholar search were excluded. The analysis yielded 103 articles: 41 by Scopus, Web of Science, and TRID and 62 by Google Scholar. The screening phase involved two different steps: The first aimed to exclude non-English documents, which is the dominant language in scientific research on this topic; the second involved reading and excluding articles not in line with the research topic. In detail, one and 87 items, respectively, were excluded during the two steps of the screening phase. The selection consisted of 16 articles, 10 from Scopus and five from Google Scholar; the literary analysis was extended using the snowball search method [36]. This search identified among the bibliography of selected documents an additional 16 documents related to the research topic. The presented methodology identified a selection of 31 articles on the topic. These were categorized according to the main research areas; for each, the temporal evolution of scientific interest was explored, reporting the main results.

## 3. Results

The classification of events provided by Allen et al. (2005), Getz (2008) [8,9] distinguish events on the basis of the number of participants (which determines the size of the event), the type of occurrence, the type of event (public or private), and the type of celebration (cultural, political, related to arts and entertainment, business and trade, educational and scientific, recreational, and sports competition). These events have particularly different characteristics in terms of demand, supply, scale, and type of investment. Transportation plays a key role in the success of events of all sizes: from small local festivals to international mega-events. According to Kassens (2010) [32], it is essential for participants and clients to be able to travel to and from an event for it to be successful; yet, transportation has not emerged as a core focus in literature. Indeed, in the reviewed literature, no rigorous approach was found regarding the identification and implementation of sustainable mobility best practices in the different cases. In fact, in many studies [5,20,25,30,37], it was noted that the identification and implementation of best practices was entrusted to the experts

themselves, who base decisions on experience. Different approaches and different points of view were developed in the papers; in fact, no established scientific methodology for making mobility sustainable for events of any type or size was found in literature. Most of the interest was focused on mega sporting events [6,7,24–28,30–32,38–48], as suggested by the number of the articles on this topic; however, the increasing number of events and event participants requires redefining and extending the concept of sustainable mobility to smaller cases where ad hoc mobility studies or transportation infrastructure cannot be built [4]. The interest in mega-events lies in the challenges they pose, and the goal of sustainable transportation for them is to ensure high standards of transportation during the event and to minimize the impact in terms of emissions and congestion, by defining good practices [6,12,31]. Another relevant consideration in the literature concerns the equity of redistribution of accessibility as a legacy of the event. Generally, studies are focused on large events because of the complexity of the challenges and relevance, and there is not a strong interest in studying smaller cases, for which best practices are supposed to be valid. In detail, the debate on event sustainability started regarding mega-events, in detail with the 1992 Rio de Janeiro Summit on Environment and Development, where the importance of sustainability (for Olympic events) in organizing the games was affirmed. The strategic importance of the transportation sector was confirmed with the 1992 Barcelona Olympics, which transformed the city by providing an extensive and reliable transportation system. Beginning with Sydney 2000, sustainability actions began to be introduced, and, in 2008 with the Beijing Olympics, efforts to improve air quality before the event were made. The turning point in terms of sustainability of mega-events occurred in 2010 for the World Cup in South Africa; nevertheless, ambitions and short timeframes led to a failure with a huge unplanned economic outlay. Moreover, in 2010 with the Commonwealth Games, interest was shown in integrating sustainability into programs by defining the green games logo while in the Vancouver Winter Games; the goal of sustainability was pursued especially in the building sector (LEED certification). Therefore, as observed by Kassens (2009) [6], much of the literature did not proactively research this topic, and the notion of sustainability appeared after its integration in major events. In fact, the interest of the scientific community toward sustainable mobility for events is recent and not yet fully established. Figure 2 shows that the first papers date back to 2007, intensifying from 2015 onward. It is interesting that, in 2020, publications were absent, which can be correlated with the absence of events for the global COVID-19 pandemic and the resulting restrictions imposed by countries. In recent years, scientific activity seems to have intensified.

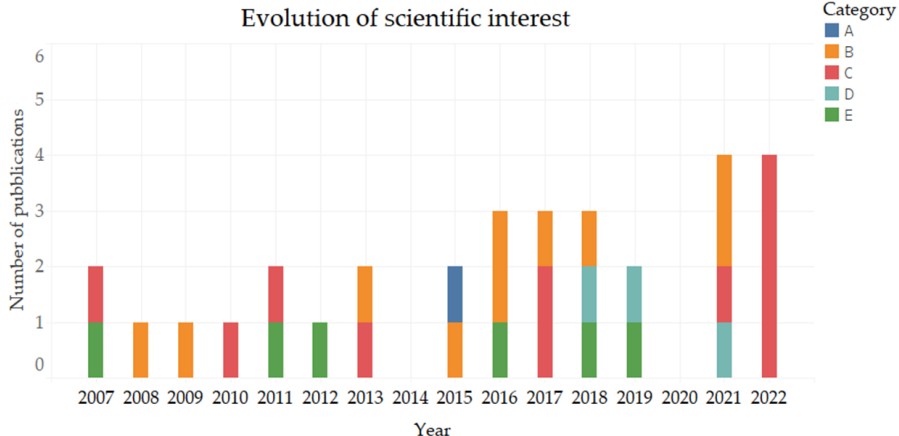

**Figure 2.** Time evolution of scientific interest toward sustainable mobility of events.

The authors' names did not recur through the various articles, which suggests little sectorization of the topic and, therefore, points out its young and under-researched state. Only A. Collins recurred four times in the environmental impact assessments of events, but his studies were not fully focused on transportation. The most recurrent keywords in the re-

viewed articles, extracted using the VosViewer [49] application, were sustainability, environmental sustainability, sustainable transportation, and transportation. Two different cluster of words were identified in the papers: satisfaction and attitude; $CO_2$ emissions and planning. This confirms that modeling is addressed primarily through behavioral models that support planning to reduce emissions. A classification of the reviewed articles is introduced to efficiently describe their added value and identify the main research directions in the area of sustainability of mobility in events. The research areas in sustainable transportation for events cover multiple interrelated areas: behavioral [19,22,50], transportation analysis and models [20,21,24,26,30–32,40,46,48,51], environmental [11,15,17,18,38,39,41,43,52–54], analysis of past case studies [25,27,28,44,47,55], and promoting guidelines [56]. These documents deal with sustainable mobility for events; however, the different focuses allow classifying them into five different areas:

A.    Promotion of guidelines and best practices,
B.    Environmental analysis and models,
C.    Transportation analysis and models,
D.    Behavioral analysis,
E.    Analysis of previous cases and their legacy.

The documents selected by the procedure are summarized in Appendix A, summarizing main information about each paper such as category, title, authors, year of publication, reference number, and a short description of the results. Below, some specifics about the classification of documents are given.

-    Documents belonging to type A deal with sustainability of mobility for events with the purpose of returning best practices or guidelines applicable to any type of event;
-    Documents belonging to type B involve the evaluation or development of methods for assessing the environmental impact of major events and related transportation;
-    Documents belonging to type C concern the analysis or interventions of purely transportation issues and interventions in the context of sustainability of large events that are hardly applicable to small events;
-    Documents belonging to type D involve behavioral studies useful for modeling the field of sustainable transportation in major events;
-    Documents belonging to type E describe, contextualize, and analyze design choices and the legacy of previous case of sustainable transportation for major events.

The most frequently discussed topics in the literature are type B and C; in detail: 6% of items fall under type A, 31% of items fall under type B, 35% of items fall under type C, 9% of items belong to type D, and 19% of items belong to type E (Figure 3).

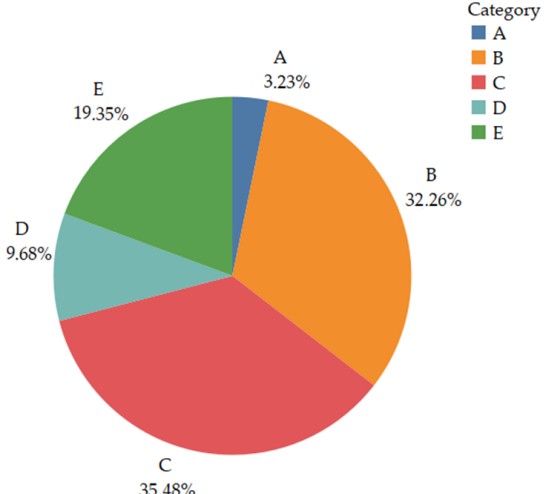

**Figure 3.** Distribution by category of the analyzed papers dealing with sustainable mobility of events.

### 3.1. Group A

As shown in Figure 3, the topic of guidelines and best practices was the least frequent topic. The first articles in the area were from 2015 and there are no successive publications in the scientific literature. The best practices approach to achieve event sustainability recurs in some guidelines [12,14]. Therefore, different topics are covered such as waste, energy efficiency, water conservation, and transportation. Focusing on the latter, the analyses behind the best practices appear coarse and shallow. Pereira's (2018) [27] approach identified sustainability practices on the basis of the opinion, acceptance, and feasibility estimated by participants and organizers by filling out questionnaires. The proposed sustainable mobility best practices for events are not based on studies that estimated and returned the environmental benefit associated with them. In particular, the transportation sector appears to be one of the most impactful and for which the following actions were proposed: incentivizing mass transport for game attendees, promoting biking, locating the event near hotels, providing electric vehicle rentals, and incentivizing carpooling and ridesharing.

### 3.2. Group B

This group, second in terms of the number of publications (Figure 3), highlights the high attention of public and private agencies to the sustainability performance of events [43]. Articles on the topic of sustainable mobility for events started from 2008 and remained almost constant until 2021. The first one was related to the major event of the FA Cup final. Mainly major events were discussed; in fact, the assessment issue was particularly deepened with regard to mega-events such as the Olympic Games and FIFA World Cup [39]. Major events are becoming increasingly common and are correlated to increased $CO_2$ emissions responsible for climate change. Therefore, it is important to reduce their impact by planning events and festivals in a more sustainable and environmentally responsible way. This has led to several developments in recent years, including the development of an International Standard ISO 20121 for Sustainable Event Management (i.e., financially viable, socially responsible, and reducing event environmental footprints), the establishment of organizations such as A Greener Festival and the Sustainable Event Alliance, and the introduction of software tools including Julie's Bicycle Creative IG (Industry Greening) Tools and other carbon calculators [18]. Therefore, the effort to quantitatively measure the environmental impact of events is also increasing [54]. However, despite event sponsors often drawing attention to the importance of environmental and socioeconomic legacy components, the environmental impacts of events are difficult to be quantitatively assessed, being complex and often occurring over extended periods [39]. In fact, despite this increasing interest toward improving the sustainability of events and festivals, there is currently a lack of agreement among methods to evaluate sustainability of transport [18]. Hanandeh (2013), Edwards et at. (2016), Triantafyllidis et al. (2018), and Cooper et al. (2021) [15,17,39,51] evaluated only the carbon footprint, while Edwards et al. (2016) [54] introduced the LCA for measuring and reducing the greenhouse gas emissions and [17] estimated the $CO_2$ emissions generated by spectator transportation using GREET 2016. Collins et al. (2009, 2017) [18,39] used the ecological footprint, whereas Collins et al. (2009) [37] analyzed the approach using a model called environmental input/output (ENVIO). Scrucca et al. (2016) [43] proposed a qualitative/quantitative method based on the EBI 2012 standard that proposes the quantitative calculation of environmental impact, commonly used to assess the sustainability of a process (or, in this case, an event), integrated into a qualitative analysis of the activities implemented in the planning and management phases of the event. According to Kwang (2011) [30], a clear environmental problem is caused by spectators traveling to the event even on a national scale. Organizers should have solid travel plans and educate spectators to employ more sustainable travel behavior. However, in support of this, according to the literature review, there is a significant lack of objective information and data, as well as appropriate scientific studies.

### 3.3. Group C

This group was the largest and contained articles with a focus on transportation models for events. The first articles date back to 2007, and publications remained almost constant until 2022 when the greatest peak occurred. This highlights the scientific interest in the topic probably driven by the ambitious and necessary sustainability goals. This has been included in the official claims of mega sporting events; however, there is no agreement in the literature on how to achieve environmental stewardship and a prolonged legacy [31]. Historically, the Games represent a big boost for the host city: new infrastructures, sports facilities, new housing and districts, etc. Unfortunately, once the event is over, it may happen that all these novelties are not always reused and are abandoned, which is the opposite of the principle of sustainable development [40]. In this sustainable development, the field of transport has a significant role. Therefore, events generate a significant carbon footprint where the transportation of participants and visitors makes the largest contribution [46]. The challenges that transport planners face are complex and derive from the high pressure that the local transport system is subject to, due to travels of regular commuters and time- and place-specific peaks of additional event demand [26]. Information technologies improve transportation system efficiency by informing users about the shortest route, but they do not incentivize the managing of travelers to achieve collective benefits in travel times and emissions [57]. In summary, the mobility problem concerns three time periods: The planning (including the candidature and the preparation processes), the delivery, and the legacy [26]. To enhance environmental sustainability in the planning phase, the facility location problem (FLP) was used by Pereira et al. (2017) [46] to determine the best location for the installation of facilities to provide for existing or envisaged consumer demand. Hosting events on a regional scale makes it possible to strengthen the interconnection and territorial cohesion, in line with the sustainability concerns, and the spatially dispersed demand should reduce the pressure on infrastructure capacity [26]. The models demand several assumptions which are often in constant evolution; hence, the versatility of the parameters is a very important point to consider. On the other hand, the background demand prediction is a significant challenge because it differs from what is usually applied [40]. A detailed action plan is required to achieve sustainable transportation through mobility and accessibility planning. The attainment of this goal requires that visitors avoid using cars to access the event and requires a strong improvement in public transportation to accommodate potential car users. The event site should also be designed to encourage the use of environmentally friendly modes of transportation, such as cars, electric buses, and bicycles [30]. Combustion technologies used are also crucial to achieve sustainability goals. A hybrid, multi-regional input/output (MRIO)-based life cycle sustainability assessment (LCSA) model was presented by Elagouz et al. (2022) [31] to classify three bus types: compressed natural gas (CNG), electric buses (EBs), and diesel buses (DBs). Even though the economic benefits of EBs are higher than those of CNG buses, with almost similar social benefits, the environmental impact of CNG buses is much less than that of DBs [31]. The successful development of sustainable mobility for mega-events must carefully align with local attitudes and behaviors. Research on mobility and travel behavior at mega-events has yet to reach its maturity, and there is a general lack of quantitative studies [24]. Therefore, information technologies also enable pursuing sustainability by saving collective travel time for users by suggesting new shorter route alternatives [57]. In fact, the whole population may be more willing to adopt recommendations during the event phase because of the high traffic inconveniences [48]. This can be achieved using big data in intelligent transportation systems, as well as by developing interactive platforms and demand services which consider featuring schedule, evacuation, sustainability, resource optimization, and environmental and economic efficiencies [21]. The number of events continues to grow, generating a real opportunity to integrate transport and cutting-edge technologies or models into the planning and regulation of new events and venues. The transport to and from those events must take a much more central role in their planning, execution, and regulation [20]. Organizations and venues can align their environmental

goals with the surrounding city's environmental objectives [52]. It was generally found that planners, governments, and residents intended to use the events as a catalyst for lasting transport improvements and achieve more sustainability promoting public transport [32].

### 3.4. Group D

Group D focused on behavioral analysis and turned out to be the second smallest. The main topics concerned the analysis of modal choices and travelers' perceived satisfaction. Curiously, accessibility to major sporting events was addressed marginally, despite the fact that transportation must be inclusive and without increasing social inequality to be sustainable. However, given the importance of the economic budget for hosting major events, it is unsurprising that enhancing accessibility has a lower priority, unless it brings the promise of market return [19]. Furthermore, transportation service quality and the physical environment of transportation significantly influence satisfaction and, in turn, positively influence users' attitudes toward sustainable choices. Identifying the significant transportation attributes and offering a high service quality are necessary to ensure satisfaction. Transportation service features actively influence users' behavior, as well as the physical environment of transportation, such as platforms and transportation vehicles, all of which have significant influences on users' satisfaction and on their choices. This can provide valuable information for the host city of events to ensure that service quality meets or exceeds passenger expectations [52]. Therefore, improving the aesthetic features of public transportation and of the destination, strengthening the connection between the local community and the event spectators, and improving the availability of information on transport services can increase spectator commitment to more sustainable modes of transport [50].

### 3.5. Group E

This group was the third largest in terms of the number of publications, which started in 2007 and discontinuously continued until 2019; the studies were usually published in the same years or in the years following major events. In general, the analysis of transportation applications and their legacy was carried out from social, performance, economic, and behavioral perspectives. The empirical findings are valuable because some practices can be generalized and deployed [25]. Green events have become popular, but few critical examinations of these events have been conducted. Much effort has been made to improve public transport efficiency and achieve an environmentally friendly transportation services; in fact, a mega-event usually catalyzes the transportation network of the host city and allows applying a sustainable transport strategy to achieve a more efficient and sustainable transport system. This goal is achieved through the planning of mass transit modal share at venues, encouraging the use of high-capacity mass transit services, and supplementing reliable bus services that leads users to mass transportation [28]. Nevertheless, large transportation investments must be commensurate with the demand, but only larger urban areas have the population needed to valorize transportation investments [47]. Indeed, in the case of expansion of public transport infrastructure, it is necessary that this should not be followed by cuts in service levels to rationalize the transportation system. This would reduce accessibility benefits, reinforcing patterns of urban inequality [27]. Then, green event management involves modeling the environmental, economic, social, and governance aspects from the host city's perspective; in addition, the investment should fulfill the organizers' intentions by leaving viable and worthwhile long-term legacies [47]. Collins et al. (2012) identified the specific consumption activities in the Tour de France in 2007 that made the most significant contribution to the overall environmental and economic impact. This provides useful information and highlights how travel behavior strongly characterizes the event sustainability. To assess the environmental impacts of the event, ENVIO and EF are valid to support policymakers in monitoring and evaluating event and post-event outcomes. This type of indicator could be useful in informing future event

planning to mitigate negative environmental consequences, particularly those linked to different travel modes [55].

## 4. Conclusions

This study presented a systematic literature review in the field of sustainable mobility for events. This research topic is a complex and recent multidisciplinary subject ranging from transportation engineering to humanities. Different authors tackled this problem from various points of view, and they focused on various aspects of the problem. In detail, these were characterized in several classes: promotion of guidelines and best practices, environmental analysis and models, transportation analysis and models, behavioral analysis, and analysis of previous cases and their legacy. The network of authors who have published articles on this topic varies, with recurrence only seen in the class "environmental analysis and models". Often, event sustainability was considered by evaluating the legacy of large events from economic, social, and environmental perspectives; however, a distinction between the different types of events and the related transport characteristics was not explored in depth. Emphasis was placed on the political and social context in which the event takes place, sustainability actions, and effectiveness during and after the event. In this context, transportation infrastructure and mobility play a key role among stakeholders because legacy is a strong lever for event acceptance by the local community. The high visibility offered by major events and strong economic interests push the countries to compete in order to host the events within their borders. There is no agreement in the literature on the environmental key indicators to be used, as with the approach to design sustainable mobility interventions for a large event; rather, experts themselves tailor their expertise and the approach to the specific case. Most of the literature underlines that the topic of sustainability of event mobility is a current and growing one. It focuses on assessing and transmitting the legacy of the event but does not provide quantitative elements to carry out this process. Therefore, there is a need for the development of scientific papers that provide quantitative evidence from simulation and empirical assessment that validates the effectiveness sustainability of interventions. In addition, it is unclear how to achieve sustainability of transportation legacy due to the bidding of cities for mega events. In this case, the lack of post event monitoring studies does not allow quantifying the ecological impact by oversized projects. In fact, it recurs in the literature that the high design standards associated with the event and the short timeframe often caused an oversizing of the post-event transportation system, followed by cuts in service levels that increased inconvenience and social inequality. Therefore, research directions must fill the gap providing quantitative data for various types of events distinguished by size, recurrence, duration, and type of celebration, by providing indications regarding the characteristics of the transportation system investigated and of the effects produced by the sustainable practices related to their cost. In this context, the development of specific indicators and dedicated procedures for the event mobility homogenizes the assessment processes. This makes comparable the results of different case studies and increases effectiveness in the knowledge transmission process. The most established approaches involve carbon footprint, ecological footprint, and life cycle analyses for which the bounds of analysis are not clearly defined. Event sustainable mobility research is limited by the paucity of quantitative studies to generalize best practices for each type of event; therefore, this work summarizes the gaps in the literature and identifies the main research directions:

- Multimodal transport modeling for large events and environmental KPI;
- Evaluation of the effect of information technology on modal and route choice;
- Validation of events traffic models;
- Assessing benefits due to integration of big data in the traffic event design and management for events.

**Author Contributions:** Conceptualization, D.B. and S.M.P.; investigation, D.B.; methodology, F.A., S.M.P. and D.B.; supervision, F.A. and S.M.P.; writing—original draft, D.B.; writing—review and editing, F.A., S.M.P. and D.B. All authors have read and agreed to the published version of the manuscript.

**Funding:** This research received no external funding.

**Institutional Review Board Statement:** Not applicable.

**Informed Consent Statement:** Not applicable.

**Data Availability Statement:** Not applicable.

**Acknowledgments:** This study was performed thanks to the Italian PON PhD program on green related topics: "Valutazione e messa a punto di buone pratiche di mobilità sostenibile per una società sportiva".

**Conflicts of Interest:** The authors declare no conflict of interest.

**Appendix A**

**Table A1.** Details of documents selected by procedure.

| Reference | Category | Title | Main Topic |
|---|---|---|---|
| [43] Scrucca et al. (2016) | B | A new method to assess the sustainability performance of events: Application to the 2014 World Orienteering Championship | Quali-quantitative method developed to measure the sustainability of events |
| [25] Wang et al. (2019) | E | An examination of a city greening mega-event | Green practices from bidding to preparing to hosting were identified and analysed |
| [21] E. Felemban, F. Ur Rehman (2022) | C | An Interactive Analysis Platform for Bus Movement: A Case Study of One of the World's Largest Annual Gathering | A big data study of GPS traces of over 17000 buses during Hajj |
| [46] Pereira et al. (2017) | C | Applying the facility location problem model for selection of more climate benign mega sporting event hosts: A case of the FIFA World Cups | A logistic approach to achieve sustainability in event design |
| [39] Collins et al. (2009) | B | Assessing the environmental impacts of mega sporting events: Two options? | Examines two approaches for quantitative impact assessment of environmental externalities connected with visitation at sporting events |
| [15] J.A. Cooper, B. P. McCullough (2021) | B | Bracketing sustainability: Carbon footprinting March Madness to rethink sustainable tourism approaches and measurements | A valuable linear model for estimating carbon footprint used for travel of fans and teams |
| [17] Triantafyllidis et al. (2018) | B | Carbon Dioxide Emissions of Spectators' Transportation in Collegiate Sporting Events: Comparing On-Campus and Off-Campus Stadium Locations | Analysis of emissions related to different modality of spectator's transportation to on-and off-campus locations |

**Table A2.** Details of documents selected by procedure.

| Reference | Category | Title | Main Topic |
|---|---|---|---|
| [48] Y. Xu, M. C. González (2017) | C | Collective benefits in traffic during mega events via the use of information technologies | Integration of different kind of data to estimate the impact of events on traffic and propose target strategies for collective good at the urban scale (Olympic Games in Rio de Janeiro) |
| [51] McCullough et al. (2022) | C | Distance decay and public transportation usage among select professional Seattle sport fans | Identify the distance from the station that makes fans dependent on another form of transportation based on the city capacity that could help sport venues fulfil their environmental goals |
| [55] Collins et al. (2012) | E | Environmental Consequences of Tourism Consumption at Major Events: An Analysis of the UK Stages of the 2007 Tour de France | Analysis of two related methods (monitoring and evaluation) for estimating the environmental effects of tourism associated with major events |
| [56] J. Casper, M. Pfahl (2015) | A | Environmental sustainability practices in U.S. NCAA Division III athletics departments | Analysis of environmental sustainability practices, strategies, and perspectives for two different cases |
| [30] K.S. Kwang (2011) | C | Exploring transportation planning issues during the preparations for EXPO 2012 Yeosu Korea | Low carbon green growth policy initiatives in the planning and operation of the Expo like limiting automobile access to the Expo site while ensuring that competitive public transportation systems and services are available |
| [52] Dosumu et al. (2017) | B | Greenhouse gas emissions as a result of spectators travelling to football in England | Calculates greenhouse gas emissions from spectator travel to and from soccer matches of different soccer levels finding that there are significant differences between them |
| [22] Mu-Chen et al. (2019) | D | How transportation service quality drives public attitude and image of a sustainable city: Satisfaction as a mediator and involvement as a moderator | Analysis of the link between transportation service quality and its impact on satisfaction as well as attitude toward and image of a sustainable city |
| [19] McGillivray et al. (2018) | D | Major sporting events and geographies of disability | Analysis of the value of major sporting events in delivering urban accessibility improvements |

**Table A3.** Details of documents selected by procedure.

| Reference | Category | Title | Main Topic |
|---|---|---|---|
| [18] A. Collins, C. Cooper (2015) | B | Measuring and managing the environmental impact of festivals: The contribution of the Ecological Footprint | Describes how the Ecological Footprint was used to calculate the environmental impact of visitors and demonstrates that this analysis can provide valuable information for festival organisers and policymakers on factors influencing the scale of a festivals' environmental impact, and the types of strategies needed to reduce the effect of visitor travel |
| [38] A. Collins, A. Flynn (2008) | B | Measuring the Environmental Sustainability of a Major Sporting Event: A Case Study of the FA Cup Final | Describes how the Ecological Footprint was used to calculate the environmental impact of the event. Provides valuable insights into the global environmental impacts generated by visitor consumption patterns and it can support policymakers and event organizers in staging sustainable events through the development of policy scenarios |

**Table A3.** *Cont.*

| Reference | Category | Title | Main Topic |
|---|---|---|---|
| [44] Lindau et al. (2016) | E | Mega events and the transformation of Rio de Janeiro into a mass-transit city | Analysis of Rio de Janeiro transport bid for the 2016 Olympic Games |
| [47] S. Essex, B. Chakley (2007) | E | Mega-sporting events in urban and regional policy: A history of the Winter Olympics | Comparative analysis of the infrastructural implications of all previous Winter Olympic Games. A key conclusion is that host cities need to have carefully integrated and realistic long-term strategies for all aspects of event-related development |
| [20] Robbin et al. (2007) | C | Planning transport for special events: A conceptual framework and future agenda for research | Describe critical issues related to event typology, destination geography to aid planning decisions |
| [41] A.E. Hanandeh (2013) | B | Quantifying the carbon footprint of religious tourism: The case of Hajj | Life cycle methodology to assess the Global Warming Potential (GWP) from the main activities of the event |
| [31] Elagouz et al. (2022) | C | Rethinking mobility strategies for mega-sporting events: A global multiregional input-output-based hybrid life cycle sustainability assessment of alternative fuel bus technologies | This article quantifies and analyses the environmental, social, and economic impacts of alternative fuel buses throughout their entire life cycle stages to support the thought process of sustainable mobility practices in hosting mega-events. A hybrid, multi-regional Input-Output based life cycle sustainability assessment model is developed |
| [40] A. Dumont (2021) | C | Sustainable travel during the Olympic and Paralympic Games: A methodology to model public transport travel for Paris 2024 | Transport modelling for a mega event |

**Table A4.** Details of documents selected by procedure.

| Reference | Category | Title | Main Topic |
|---|---|---|---|
| [32] E. Kassens-Noor (2010) | C | Sustaining the Momentum: Olympics as Potential Catalyst for Enhancing Urban Transport | Analysis of four cases: The challenges the Olympics impose on urban transport systems, the long-term transport goals each city intended to achieve, how the cities changed and which of the measures that supported the passenger operations were sustained |
| [26] Lopes dos Santos et al. (2022) | C | The Long-Standing Issue of Mobility at the Olympics: From Host Cities to Host Regions in the Ongoing Case Study of Milan–Cortina 2026 | Identify and discuss new challenges that the possibility of regions being hosts bring for mega-event mobility planning |
| [54] Edwards et al. (2016) | B | The methodology and results of using life cycle assessment to measure and reduce the greenhouse gas emissions footprint of "Major Events" at the University of Arizona | Life cycle assessments of two events, analysis of each category of impact and definition of best practices for mitigating the impact of events on a category-by-category basis |
| [11] Burton et al. (2021) | B | The relationship of climate change & major events in Austria | The paper discusses the relationship between events and climate change by looking at mitigation measures on the one hand as well as adaptive strategies taken by event organizers on the other |
| [28] M. Craig (2011) | E | TransLink and the 2010 Olympic Winter Games | Description of the TransLink transportation network planning for 2010 Olympic and Paralympic Winter Games |

**Table A4.** *Cont.*

| Reference | Category | Title | Main Topic |
|---|---|---|---|
| [27] R. H.M. Pereira (2018) | E | Transport legacy of mega-events and the redistribution of accessibility to urban destinations | Delimitation of transport legacies and its social impacts in terms of how such developments can reshape urban accessibility to opportunities in Rio de Janeiro (Brazil) in preparation for the 2014 World Cup and the 2016 Olympic Games |
| [24] A. Malhado, R. Rothfuss (2013) | C | Transporting 2014 FIFA World Cup to sustainability: Exploring residents' and tourists' attitudes and behaviours | Surveying attitudes and travel behaviour of tourists and residents for a mega event and demonstration that successful of sustainable mobility for mega-events must align with local attitudes and behaviours |
| [50] Martins et al. (2021) | D | Understanding spectator sustainable transportation intentions in international sport tourism events | Investigation that aims to identify which factors influence spectator sustainable transport intentions, performing a segmentation based on their responses to define transport strategies |

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
