# Peer review of "Sustainable Transportation for Events: A Systematic Review"

_sustainability, doi:10.3390/su142315815_

Round 1
Reviewer 1 Report
First of all, I appreciate the opportunity to review your paper Sustainable transportation for events: A systematic review. The topic is actual and important. Suggestions for paper improvement:
· The abstract is not well written. More findings must be highlighted. Bullets and numbering in the abstract are not common (lines 15-18).
· Keywords should include review, literature review, or SLR.
· It is necessary to understand the purpose and aim of the paper as well as its "position" in relation to previous research (gap analysis). This is partially done in the introduction.
· There is no strong SLR methodology. This kind of paper must have very clear methodology with a methodology diagram. There are numerous papers for methodology (see Suggested References).
· The review paper should not just be a list of what everyone has done but should identify trends and gaps in the literature and offer suggestions for furthering the field relative to the specific phenomenon, with a VERY STRONG CRITICAL VIEW AND VERY STRONG METHODOLOGY
· Conclusion section is not on a satisfactory level. Clearly state your unique research contributions in the conclusion section.
· Limitations of this research are missing.
· Future research directions in conclusions should be stronger.
· Table 1 and Appendix A are not well presented (especially Appendix)
· Scientific and practical contributions should be emphasized.
Suggested References
Denyer, D. & Tranfield, D., (2009). Producing a systematic review. In D. Buchanan & A. Bryman (eds.) The sage handbook of organizational research methods. Sage Publications Inc., Thousand Oaks, CA, 671-689.
Kilibarda, M., Andrejić, M., & Popović, V. (2020). Research in logistics service quality: a systematic literature review. Transport, 35 (2), 224-235.
Author Response
Title: Sustainable transportation for events: A systematic review
Manuscript ID: sustainability-2035858
First of all, we would like to thank the Editor and the two Reviewers for their positive comments and relevant suggestions that helped us improving the paper. We strongly hope that the changes brought to the paper are appropriate to consider it for publication. We addressed each single point raised by the Reviewers, as described below.
Reviewer 1
- The abstract is not well written. More findings must be highlighted. Bullets and numbering in the abstract are not common (lines 15-18).
We have received the comment and improved abstract by removing bullets, numbers and including more findings and limits.
- Keywords should include review, literature review, or SLR.
We included among the keywords “literature review”.
- It is necessary to understand the purpose and aim of the paper as well as its "position" in relation to previous research (gap analysis). This is partially done in the introduction.
We highlighted the literature gaps about the topic and the RQ to clarify the purpose of the paper. The analysis of the gaps in the literature was deepened.
- There is no strong SLR methodology. This kind of paper must have very clear methodology with a methodology diagram. There are numerous papers for methodology (see Suggested References). Suggested References:
Denyer, D. & Tranfield, D., (2009). Producing a systematic review. In D. Buchanan & A. Bryman (eds.) The sage handbook of organizational research methods. Sage Publications Inc., Thousand Oaks, CA, 671-689.
Kilibarda, M., Andrejić, M., & Popović, V. (2020). Research in logistics service quality: a systematic literature review. Transport, 35 (2), 224-235.
The description of the methodology was strengthened including the suggested documents.
- The review paper should not just be a list of what everyone has done but should identify trends and gaps in the literature and offer suggestions for furthering the field relative to the specific phenomenon, with a VERY STRONG CRITICAL VIEW AND VERY STRONG METHODOLOGY
The description of trends, gaps and suggestions for future research was included.
- Conclusion section is not on a satisfactory level. Clearly state your unique research contributions in the conclusion section.
- Limitations of this research are missing
- Future research directions in conclusions should be stronger.
- Table 1 and Appendix A are not well presented (especially Appendix)
- Scientific and practical contributions should be emphasized.
The conclusions included research limitations and directions. Scientific contributions were emphasized, and tables presentation was more detailed.
Reviewer 2 Report
This paper gives a systematic review for sustainable transport event, the comments are as follws.
(1) What does 'event' stand for? The authors should give the explanation in the paper.
(2) There are many kinds of transportation modes, what is the sustainable transportation in your paper?
(3) As a review paper, I find the recent three year paper is less. In recent years, there is a large number of related papers in sustainable transportation. I suggestion the authors add some of them.
(4) The format of paper should be better edited.
Author Response
Title: Sustainable transportation for events: A systematic review
Manuscript ID: sustainability-2035858
First of all, we would like to thank the Editor and the two Reviewers for their positive comments and relevant suggestions that helped us improving the paper. We strongly hope that the changes brought to the paper are appropriate to consider it for publication. We addressed each single point raised by the Reviewers, as described below.
- What does 'event' stand for? The authors should give the explanation in the paper.
A brief definition of events was added in the introduction section including relevant references
- There are many kinds of transportation modes, what is the sustainable transportation in your paper?
As specified the introduction, our study focus on promising sustainable transport modes and technologies, such as: public transport, active mobility, on-demand services, and intelligent transportation system.
- As a review paper, I find the recent three year paper is less. In recent years, there is a large number of related papers in sustainable transportation. I suggestion the authors add some of them.
This is a targeted study that aims to probe into the transport sustainability for events and not the sustainability of transport in general. If we had considered transportation sustainability from a broader perspective, we would have lost the specific focus and uniqueness of this review.
- The format of paper should be better edited.
We have improved the format of images, tables, and captions
Round 2
Reviewer 1 Report
The paper should be accepted for publication.